# Methods for Predicting Ethylene/Cyclic Olefin Copolymerization Rates Promoted by Single-Site Metallocene: Kinetics Is the Key

**DOI:** 10.3390/polym14030459

**Published:** 2022-01-24

**Authors:** Amjad Ali, Ahmad Naveed, Tahir Rasheed, Tariq Aziz, Muhammad Imran, Ze-Kun Zhang, Muhammad Wajid Ullah, Ameer Ali Kubar, Aziz Ur Rehman, Zhiqiang Fan, Li Guo

**Affiliations:** 1Research School of Polymeric Materials, School of Materials Science & Engineering, Jiangsu University, Zhenjiang 212013, China; amjadali@zju.edu.cn (A.A.); ahmad.naveed321@gmil.com (A.N.); 2Interdisciplinary Research Center for Advanced Materials, King Fahd University of Petroleum and Minerals (KFUPM), Dhahran 31261, Saudi Arabia; 3School of Engineering Yunqi Campus, Westlake University, Hangzhou 310024, China; TARIQ@westlake.edu.cn; 4Department of Chemistry, Government College University, Lahore 54000, Pakistan; Imran.zafar87@icloud.com (M.I.); Azizryk@yahoo.com (A.U.R.); 5MOE Key Laboratory of Macromolecular Synthesis and Functionalization, Department of Polymer Science and Engineering, Zhejiang University, Hangzhou 310027, China; 21829019@zju.edu.cn (Z.-K.Z.); fanzq@zju.edu.cn (Z.F.); 6Biofuels Institute, School of Environment and Safety Engineering, Jiangsu University, Zhenjiang 212013, China; Wajid_kundi@ujs.edu.cn; 7State Key Laboratory of Clean Energy Utilization, Zhejiang University, Hangzhou 310027, China; Ameerali.kubar@yahoo.com

**Keywords:** metallocene, 4-vinyl-cyclohexene, borate, ethylene, kinetics, polymerization

## Abstract

In toluene at 50 °C, the vinyl addition polymerization of 4-vinyl-cyclohexene (VCH) comonomers with ethylene is investigated using symmetrical metallocene (*rac*-Et(Ind)_2_ZrCl_2_) combined with borate/TIBA. To demonstrate the polymerizations’ living character, cyclic VCH with linear-exocyclic**_π_** and endocyclic**_π_** bonds produces monomodal polymers, but the dispersity (Ɖ) was broader. The copolymers obtained can be dissolved in conventional organic solvent and have excellent thermal stability and crystalline temperature (Δ*H*_m_), and their melting temperature (Tm) varies from 109 to 126 °C, and Δ*H*_m_ ranges from 80 to 128 (J/g). Secondly, the distribution of polymeric catalysts engaged in polymer chain synthesis and the nature of the dormant state are two of the most essential yet fundamentally unknown aspects. Comprehensive and exhaustive kinetics of E/VCH have shown numerous different kinetic aspects that are interpreted as manifestations of polymeric catalysts or of the instability of several types of active center [Zr]/[C*] fluctuations and formation rates of chain propagation *R_p_*E, *R_p_*VCH, and propagation rate constants *k_p_*E and *k_p_*VCH, the quantitative relationship between *R_p_*E, *R_p_*VCH and *k_p_*E, *k_p_*VCH and catalyst structures, their constituent polymer Mw, and their reactivity response to the endocyclic and exocyclic bonds of VCH. The kinetic parameters *R_p_*E, *R_p_*VCH, *k_p_*E, and *k_p_*VCH, which are the apparent rates for the metallocene-catalyzed E/VCH, *R_p_*E, and *kp*E values, are much more significant than *R_p_*VCH and *kp*VCH at 120 s, *R_p_*E and *R_p_*VCH 39.63 and 0.78, and the *k_p_*E and *kp*VCH values are 6461 and 93 L/mol·s, respectively, and minor diffusion barriers are recommended in the early stages. Compared with previously reported PE, *R_p_*E and *k_p_*E values are 34.2 and 7080 L/mol·s. VCH increases the *R_p_*E in the initial stage, as we are expecting; this means that the exocyclic bond of VCH is more active at the initial level, and that the chain transfer reaction of cyclic internal π double is increased with the reaction time. The t_p_ versus *R_p_*, *k_p_*, and [Zr]/[C*] fraction count may be fitted to a model that invokes deactivation of growing polymer chains. At t_p_ 120–360 s higher, the incorporation rate of VCH suppresses E insertion, resulting in reduced molecular weight.

## 1. Introduction

Cyclic dienes (e.g., norbornane N.B), dicyclopentadiene (DCPD), 5-ethylidene-2-norbornene (ENB), 5-vinyl-2-norbornene (VNB), and 4-vinylcyclohexene (VCH) can be prepared by a reaction in multiple fundamentally different ways, relying upon the weather already bound by vinyl addition reactions, radical and cationic polymerization, and ring-opening metathesis olefin polymerization [1,2,3,4]. Compared to cyclic dienes synthesized by radical or cationic and ring-opening polymerization, vinyl addition cyclic dienes exhibit excellent thermal stability, low dielectric constant, and higher glass-transition temperature, all of which are a result of the double bond cyclic backbone chain structure [5,6,7]. Co- and terpolymers of ethylene/propylene (EPM) and ethylene/propylene/diene (EPDM) are significant manufacturing thermoplastics [8,9,10,11]. Throughout the last few decades, EPM and EPDM have been suitable choices for synthetic plastic and rubber due to their superior weatherability and mechanical properties. The α-olefin (propylene, cyclic and linear diene)-based polymers play a significant part in people’s daily lives. A substantial amount of rubber, plastics, and fiber materials are manufactured each year using these monomers. According to Worldwide Information, Inc.’s analytics, EPDM global production is growing at a rate of 4% per year and will exceed 1.9 million tons annually by 2023 [10,12,13,14,15].

Typical slightly elevated catalysts are based on metallocene or post-metallocene complexes. Remarkably, metallocene catalysts include sandwiches having C_1_ [8,9], C_s_ [9,10], and C_2_ [11,12] symmetries, as well as complexes with restricted geometry [13]. Half-sandwich Ti (IV) complexes containing nitrogen anionic donor ligands such as ketimides, phosphine imides, iminoimidazolidinates, amidinates, and guanidinates are especially promising post-metallocene catalysts for EPDM manufacture [14]. Since the late 1980s, a wide range of single-sight ansa-metallocene polymeric catalysts have made it possible to produce EPM and EPDM with a wide range of composition, an extensive range of physical properties, structural diversity, greater incorporation of numerous cyclic and linear dienes without gel growth, and the production of a higher molecular weight EPM and EPDM with a narrow molecular weight distribution at elevated pressure and temperatures [14,16,17,18]. The single-site Group IV metallocene catalysts discovered in the late 1950s have received considerable commercial attention. One of the essential characteristics of these catalysts is their open active site, which enables the incorporation of additional olefins or α-olefins into P.E. A number of papers in the literature describe the co- and ter- polymerization of ethylene with linear α-olefins such as hexene and octane, as well as cyclic monomers such as norbornene (N.B., ENB, VNB, and DCPD) and VCH. In addition, recent advances in metallocene-catalyzed chain-shuttling polymerization have helped broaden the spectrum of these already diverse polymeric catalyst systems [19,20,21]. Similarly, the manufacturing output of P.E., P.P., EPM, and EPDM polymers with conventional Ziegler–Natta polymeric catalysts has decreased relative to polyolefin manufacture utilizing metallocene [17,18,19,20,21,22].

In post-metallocene and metallocene catalysts, the most efficient cocatalysts are methylaluminoxane and their derivatives methylaluminoxane (MAO) and modified methylaluminoxane (MMAO), which are employed in the commercial synthesis of EPM and EPDM [22,23,24]. Regardless of the fact that MAO is, in actuality, the facilitator of such systems, its use as an activator has considerable disadvantages. These include high cost, the requirement to employ it in large molar excess to pre-catalysts, and the loss of stability during prolonged storage, resulting in loss of activating capabilities and irreproducibility of polymers properties. According to the literature, triisobutylaluminium (TIBA)/borate is an excellent activator of metallocene complexes in the polymerization of olefins [25,26,27,28].

Metallocenes/TIBA/borates and metallocenes/triethylaluminium (TEA)/borates have been exploited as catalysts for both homo-, co-, and ter- polymerizations. While these catalytic systems have demonstrated impressive outcomes in homopolymerization, several complications have arisen when two or more monomers are supplied in the system for polymerization. Depth of knowledge of the kinetics and mechanism of polymeric metallocene canalization α-olefin co/terpolymer process would impressively aid the development of better polymerization process optimizations and an improved metallocene catalyst system. A kinetic investigation based on active center [Zr][C*] concentration measurements during the polymerization process can offer direct characteristic information on polymeric catalyst activation efficiency and the nature of the active sites, which are crucial for illustrating the catalysis mechanism. There are only a few publications in the literature on the kinetics and mechanism of metallocene-catalyzed olefin co- and ter- polymerization based on active center counts. Furthermore, the contributions of R. Landis [25], Chen [4], Fan [22], Kaminsky [26], Bochmann [27], and M. Waymouth [5] have resulted in significant progress in the establishment of innovative copolymerization metallocene catalysts.

Unfortunately, few catalysts have been defined in terms of kinetics, partly because of the lack of procedures for required important information, such as active site [Zr][C*] counts. Although the kinetics of a wide range of catalysts have yet to be determined, the fundamental aspects of catalytic olefin polymerization are well understood [19,20,21]. There are multiple steps in *rac*-Me_2_Si(2-Me-4-Ph-Ind)2ZrCl2/[Ph_3_C][B(C_6_F_5_)_4_], *rac*Et(Ind)_2_ZrCl_2_/[Ph_3_C][B(C_6_F_5_)_4_]-catalyzed olefin polymerization: initiation into a metal alkyl (M-R) bond, propagation through the resulting inserts, and chain transfer following either 2,1 or 1,2-α-olefin inserts. This has been demonstrated in previous studies [26,29]. Catalyst inactivity or dormancy, death, and other improbable outcomes are possible extra phases [7,25]. It is currently difficult for researchers to quickly determine the kinetics of polymerization for many different catalysts. Finding new and better ways to count active sites has historically contributed significantly to the field’s core knowledge and technological advancements [21,30].

The present study’s major objective is to examine the kinetic behavior of ethylene and 4-vinyl cyclohexene (VCH) with symmetrical *rac*-Et(Ind)_2_ZrCl_2_/activated by triisobutylaluminum-/[Ph_3_C][B(C_6_F_5_)_4_] cocatalysts in gaseous phase reactors to use a model with parameters determined from gas- and slurry-phase *rac*-Et(Ind)_2_ZrCl_2_/MMAO. The number of active centers ([C]/[Zr]) in different copolymerization runs was measured using the quench-labeling technique utilizing (TPCC) as a quenching agent to better understand the process of E/P copolymerization with metallocene/borate catalysts. In order to better understand the mechanism of E/VCH with the above catalyst system, by quench-labeling polymer propagation chains in the system using thiophene-2-carbonyl chloride as the quenching agent, the practical approach adopted in the study was an experimental assessment of active center fraction (molar ratio of metallocene and active centers in the system). The propagation rate constant, *R_p_*E, and *R_p_*VCH, the kinetic rate constant of chain propagation (*k*_pE_ and *k_p_*VCH), were determined based on the data of polymerization activity and equilibrium of ethylene. Furthermore, we explored the connection between catalyst arrangement, polymer morphology, and microstructure in E/VCH copolymerization. It will be possible to learn more about the mechanism of single-site metallocene-catalyzed ethylene with α-olefins copolymerization by studying the kinetics of E/VCH, particularly the association between the catalytic active-site properties and the structure of the polymeric catalyst. It has been reported in our previous study that the active centers in *rac*-Et(Ind)_2_ZrCl_2_/MMAO and *rac*-Et(Ind)_2_ZrCl_2_/triisobutylaluminum/[Ph_3_C][B(C_6_F_5_)_4_]-catalyzed E/P copolymers grew steadily at first, but eventually reached an all-time high of 98.7%, which was significantly higher than the maximum value of polyethylene [11,25,31]. It will be much more encouraging to examine how the activation efficiency and dynamic as well as kinetic features of *rac*-Et(Ind)_2_ZrCl_2_/triisobutylaluminum/[Ph_3_C][B(C_6_F_5_)_4_] are affected by the replacement of propylene with cyclic diene and their endocyclic double bond effect on the propagation rate constant, *R_p_*E, and *R_p_*VCH, kinetic rate constant of chain propagation (*k*_p_E and *k_p_*VCH). However, active site counts (or the proportion of active catalyst) must be measured over time to establish precise polymerization kinetics.

## 2. Polymerization

All E/VCH polymerizations were carried out in a 100 mL Schlenk flask with a rounded bottom. Before beginning polymerization, the reactor was heated up to 100 °C under vacuum for 45 min and then backfilled with nitrogen gas. The Schlenk flask reactor was filled with the required volume of toluene and then saturated with 0.1 MPa ethylene at 50 °C. After injecting VCH (0.06 mol/L) into the reactor, adding TIBA (2 M) running it for 5 min, and followed by a suitable quantity of zirconocene, the catalyst and borate in toluene solutions were added. During copolymerization, 0.1 MPa E was continuously fed into the reactor. TPPC (TPCC/Al = 2) was injected five minutes after the intended time. The unreacted TPCC and TIBA decomposed with dehydrated ethanol containing 2% hydrochloric acid. This was followed by filtering of the precipitated polymers, several washes with 95% ethanol, and subsequently drying in a vacuum at room temperature to constant weight [25,26,32].

## 3. Materials

Handling moisture and air-sensitive materials were accomplished in an inert glove box. Zhejiang Mixing Gas Co. (Hangzhou, China) supplied the ethylene and further refined it with a gas purification column and molecular sieves. By flushing the gases, oxygen was removed. Comonomer 4-vinyl-cyclohexene (VCH) was obtained from Aldrich and dehydrated by stirring it for 6 h over CaH_2_. It was then refined further by dehydrating it through 4 Ă molecular sieves, distilling it under reduced pressure, and storing it in an N_2_ gas atmosphere. Sigma-Aldrich supplied the symmetrical ansa-metallocene (*rac*-Et(Ind)_2_ZrCl_2_). Albemarle Co. Shanghai, China. provided the cocatalyst triisobutylaluminum (TIBA), and the borate (Ph_3_C)B(C_6_F_5_)_4_ activator was prepared according to the published literature. J&K Scientific, PR China, supplied the quenching agent TPCC with 98% purity. Toluene reaction solvent (Aldrich, 99.7%) was refluxed for 4 h over sodium benzophenone, distilled, and stored on molecular sieves under nitrogen. The TPCC was distilled first, then diluted to 2.0 M with distilled n-heptane before use.

## 4. Characterizations

An ultra-violet fluorescence sulfur detector, the YHTS-2000, was utilized to measure the sulfur (S) content of each E/VCH polymer sample and take the average value content from pure E/VCH synthetic polymer parallel samples. Each sample was assessed for sulfur (S) content three times, and the mean values were used to calculate the S percentage.

In making the comparison with the quench-labeled pure samples, which already had S concentrations ranging from 5 to 20 ppm, the S content in a blank E/VCH sample produced under typical conditions but without the quenching step was approximately 0 ppm (see Figure 1) [29,33]. A mechanistic model was proposed to explain the observed phenomena in detail. E and VCH copolymerization create non-active sites, and the VCH endocyclic double bonds become chelated with Zr metal and interact with the catalyst ligand. Thus, the latter portion of this study explains how VCH activates the inactive site.

The resulting copolymers ^1^HNMR were obtained using a Varian Mercury plus 300 spectrometers in pulse Fourier transform type functioning at 75 MHz. The experiments were carried out in o-dichlorobenzene-d4 with a copolymer solution containing 10% by weight and hexamethyldisiloxane as an internal reference at 120 °C. Cr (acac)_3_ was introduced to moderate the relaxation duration of carbon atoms, and the pulse delay time was set to 3 s [31,34]. The pulse was angled at 90°. It took 0.8 s to complete the acquisition. The spectral width was 8000 Hz, and inverse gated decoupling was utilized for integration. On average, around 4000 scans were collected. Previous research confirmed the ^1^HNMR peak assignment [34]. The ^1^HNMR spectra of copolymer samples are shown inAppendix A. ^1^H NMR was used to calculate the amount of inserted comonomers in the copolymers and the comonomer conversion according to the literature [35,36,37].

Differential scanning calorimetry (DSC) analysis was carried out using TA Q200 equipment calibrated with indium and water. The sample, typically 3–5 mg, was employed in a sealed aluminum pan. To eliminate thermal behavior, the taster was heated to 150 °C for 5 min before being cooled at a rate of 10 °C/min to 20 °C. Lastly, the melting curve was obtained by gradually heating the sample to 180 °C at a rate of 10 °C/min [5,38].

The molecular weight (Mw) and molecular weight distribution (Mw/Mn) of E/VCH copolymers were determined using a high-temperature gel permeation-chromatography PL 220 GPC apparatus with three PL-gel 10 m MIXED-B columns and 1,2,4-trichlorobenzene as the eluent at a flow rate of 1.0 mL/min at a temperature of 150 °C. It was planned for conducting universal calibration against thin polystyrene standards.

## 5. Result and Discussion

Metallocene catalysts combined with MAO, MMAO, and TIBA cocatalyst have recently been shown to be useful polymeric catalysts in the homopolymerization of ethylene and their co-/ter- polymerization with α-olefins and cyclic olefins [29,31]. The co- and ter- polymerization of E with P and bicyclic (N.B., ENB, and VNB) olefins have been investigated using certainly known metallocene catalysts in the literature [9,12]. However, one of the recommended cyclic dienes, VCH, possesses both linear-exocyclic**_π_** and endocyclic**_π_** bonds, but they differ in terms of reactivity. Under identical experimental conditions, VCH exhibits exclusive steric hindrance and reactivity, resulting in various copolymerization characteristics. We projected that TIBA (800 Equiv. to Zr) would successfully activate this isospecific *rac*-Me2Et (Ind)_2_ZrCl_2_ catalysts in the presence of borate (2 Equiv. to Zr), which serves as an activator. In addition, TIBA was allowed to react in toluene at 50 °C for 10 min with catalyst precursor and diene impurities. The ethylene (99.9% pure) gaseous monomer was refilled during the reaction time progress by keeping a constant gas pressure (0.1 MPa). At the same time, the liquid VCH comonomers were introduced in the reactor at the start. Under the parameters used, no copolymerization of E with VCH was reported before. However, Table 1 summarizes the typical results of E/VCH copolymerizations. By increasing the polymerization time (t_p_) with a fixed feed ratio of 0.06 mol/L VCH with ethylene copolymers, up to 3% may be generated. E/VCH copolymers with a low VCH content were completely insoluble in toluene at 50 °C, indicating that they were not gel-free. Copolymers containing 2–3% VCH were anticipated to be soluble in 1,2,4-trichlorobenzene at 130 °C. Swaminathan Sivaram et al., and Fan et al., found that when the comonomer concentration of the feed increased, the catalytic activity and copolymer intrinsic viscosity was reduced [39]. Following a quick examination of the reaction conditions, it was determined that the most effective results were achieved at 50 °C and an Al/Zr ratio of 800. Earlier research exposed that the catalyst *rac*-Me_2_Et(Ind)_2_ZrCl_2_ with ethylene bridges had the highest ethylene homo- and E/P copolymerization activity [7,25,29]. TIBA immediately activated the *rac*-Me2Et(Ind)2ZrCl2/borate and the copolymerization occurred at considerable rates, providing strong ethylene/VCH copolymerization activity in the beginning. Interestingly, increasing ethylene/VCH copolymerization’s reaction time resulted in significantly reduced catalytic activity under identical parameters. As a result, we selected the quenched labeling approach to investigate how bicyclic olefins (VCH) with either an endocyclic**_π_** or an exocyclic linear**_π_** bond affect metallocene catalytic activity (Figure 1).

The impact of cyclic and bicyclic olefins exhibiting linear-exocyclic**_π_** and endocyclic**_π_** bonds on homogeneous metallocene catalysts with a MAO, MMAO catalyst system, including a metallocene/TIBA/borate system, has not been well explored. It has long been known that raising the quantity of cyclic and linear dienes in the feed, as predicted, reduces the polymerization catalyst activities, but results in higher levels than ethylene homo- and lower levels than ethylene/propylene copolymerizations. It has been shown that the activity of E/VCH copolymers decreased from the early to late-1800 s. In contrast, the active center [C*]/[Zr] fraction was more significant than ethylene homopolymerization but lower than their copolymerization with propylene, and grew dramatically in the early 1000 s. Table 1 and Figure 1 found that E homo- and E/VCH copolymerization showed higher average activity in the 1800 s. However, the time-dependent variation in the chain propagation rate (*R_p_*) was calculated by differentiating the polymer yield against the time curve (Appendix A). The ([C*]/[Zr]) of pure polymer samples was calculated using the equation [S] = [C*] and the reaction time (Appendix A).

The rate equation *R_p_* = *kp* [C*][M] ([M] = 0.085 mol/L) was used to calculate the time-dependent change of *kp*. Table 1 and Table 2 summarize the results of E/VCH, including time-dependent active center fraction [C*]/[Zr], *R_p_*, *k_p_*, Mw, MWD, and thermal characteristics (*T*_m_, Δ*H*_m_). In contrast, Figure 2 shows active center fraction variation with time and polymerization propagation rates.

Table 2 Summary of the E/VCH copolymerization observations, including the time-dependent [C*]/[Zr] ratio and kinetic data such as *R_p_*E, *R_p_*VCH, *k_p_*E, and *k_p_*VCH. Both *kp*E and *kp*VCH values dropped considerably with t_p_, as shown in Figures 7 and 8, respectively.

However, as shown in Figure 3, the initial polymerization rate of E/VCH copolymerization was even slightly higher than that of E homopolymerization but lower than that of E/P copolymerization, but later ethylene homo and E/VCH showed a more stable rate than E/P copolymerization when the reaction duration was prolonged to 1800 s. It is worth noting that both polymerization systems with the same catalyst showed a high initial polymerization rate, which can be taken as evidence for the rapid initiation of the active centers.

At 50 °C, the E/VCH polymerization activity level decreased in the reaction period of 120 s to 1800 s, and the active center fraction [Zr]/[*C] considerably increased in the initial 840 s and subsequently tended toward a steady state of 73%. Figure 2 is comparable to P.E. but lower than their E/P copolymerizations [26]. The decrease in E solubility in toluene at higher temperatures allows for greater VCH incorporations, as investigated by M. Marques and J. C. W. Chien et al. [34,40,41]. However, a significant decrease in VCH insertion is seen compared to E/NBE and E/ENB copolymerization [34,36,42]. Nevertheless, polymerization activity and molecular weight of obtained E/VCH copolymers were comparable with E/NBE and E/P/ENB copolymers. Nevertheless, propagation rate, yield, and Mw of E/NBE and E/P/ENB were much higher than E/VCH copolymers under similar conditions. In E/VCH copolymerization, VCH tends to suppress the ethylene incorporation rate, especially during 120 s to 360 s, and to lower the copolymers’ molecular weight. Comparable findings were observed in earlier investigations [34,43,44,45] Additionally, the VCH endocyclic_π_ bond may interact with the active species distantly, resulting in a smaller incorporating gap and quicker chain transfer. According to these conclusions, small changes in the architectures of comonomers and catalysts can alter polymer behavior and activity. Furthermore, active sites accumulating on catalyst surfaces account for the rapid growth in inactive sites at the introduction stage. As a result, the cocatalyst and arriving monomer may readily access these active sites, activate before contact, and begin quickly. We investigated the polyethylene chain propagation rate *R_p_* in an earlier study under identical polymerization conditions. We compared it to the E/VCH polymerization rate, finding that P.E. had a significantly higher *R_p_* at first but that the E/VCH copolymerization *R_p_* was more stable when the reaction time was extended to the 1800 s (see Figure 2). It suggests that both polymerization systems demonstrated more excellent early polymerization rates, indicating faster active center activation.

The molecular weight (Mw) of the E/VCH copolymers was lower at the beginning of the reaction but gradually rose with reaction time (t_p_). A narrow molecular weight distribution (MWD) of E/VCH copolymers suggests numerous active [Zr]/[C*] sites in the catalytic system, as the MWD was presumably more than 2 (Appendix A). The chain transfer of the active centers with TIBA was possibly responsible for the *rac*-Me_2_Et(Ind)_2_ZrCl_2_/borate system. Under comparable polymerization conditions, the *rac*-Me_2_Et(Ind)_2_ZrCl_2_ metallocene combined with MMAO produced higher molecular weight than the borate cocatalyst system but analogous MWD to the *rac*-Me_2_Et(Ind)_2_ZrCl_2_/MMAO catalyst. For the *rac*-Me_2_Et(Ind)_2_ZrCl_2_/borate system, chain transfer of the active centers with TIBA could have been responsible.

Depending on the content of dienes, the E/dienes copolymers were amorphous or had a Tm of 100–120 °C. The thermal properties of E/VCH copolymers with polymerization time are presented in Table 1. Generally, cyclic olefins are bulkier than ethylene and α-olefins and inhibit the rotational movement of polymer, cyclic olefin structure, and concentration; polymerization reaction time also substantially influences the thermal properties (crystalline temperature (Δ*H*_m_) and melting temperature (Tm). The Tm varies from 101 to 121 °C, but the Δ*Hm* varies from 9 to 16 (J/g). These results are strongly connected to the copolymer composition, indicating that composition drift may have an unexpectedly detrimental influence on thermal properties. VCH-rich segments and chains early in the process might suggest amorphous and crystalline segments with different ethylene sequences, such as higher VCH chain segments that melt at lower temperatures and lower comonomer blocks that melt at higher temperatures.

The E/VCH copolymers ^1^HNMR (Figure 4) revealed a distinct vinyl pattern between the 4.7 and 6.0 ppm endocyclic_π_ bond and the absence of a strong triplet at the 6.1 ppm linear-exocyclic_π_ bond. The selectivity of copolymerization by enchainment of the endocyclic_π_ bond was determined by ^l^H NMR using an optimal physical combination of E/VNB copolymers comprising a bicyclic olefin with either an endocyclic_π_ or a linear-exocyclic_π_ bond. ^1^HNMR was also utilized to determine comonomer content (Appendix A). In addition, Figure 5 shows the VCH mol% graph in the E/VCH copolymers vs. the [Zr]/[C*] in the system.

As we explored, the crystalline appearances of the E/VCH copolymer are lower, and the initial (120–360 s) polymerization rates (*R_p_*E and *R_p_*VVCH) are higher due to the more significant insertion VCH in the growing polymer chain; the *R_p_* reaction rates remain constant with decreasing VCH mol%. As shown in Figure 6, The VCH mol% and *R_p_*E and *R_p_* in the E/VCH copolymers are dropped as the reaction time increases from 360 s. This means that the chain termination reaction in E/VCH is faster at the later stage than the initial stage.

The results are analogous to those found in homo- and copolymerization systems of E and α-olefin. In the E/VCH copolymers, *k_p_*E values are noticeably more significant in both stages. At t*_p_* 120 s, the *kp*VCH and *k_p_*E are 93 and 6410 L/mol·s; correspondingly, in *k_p_* VCH the beginning stages, the slighter diffusion barriers near the real propagation rate constant were proposed. However, the drop in *k_p_*, as shown in Figure 6 and Figure 7, may be due to a diffusion barrier, as the rate of E diffusion from the bulk of the solution to the [C*]/[Zr] decreases rapidly when the primary consignment of polymer chains form and consolidate around the [C*]/[Zr]. Though the [C*]/[Zr] ratio was enhanced upwards of 2–3 times in the first 480 s, the potential reactivity difference among active centers activated at various phases of the reaction may also produce a substantial variation in the propagation rate constant values. For example, suppose the active catalyst site which is engaged during the 480–840 s period is 40% less active than those catalysts activated during the 120 s, and the former account for approximately 60% of all active centers. In that case, the average propagation rate constant value at 240 s polymerization time will be just 73% at 120 s polymerization time, suggesting no diffusion barrier in the system. In a previous study, we investigated the polyethylene and polypropylene homo- and ethylene/propylene copolymerization propagation rate constant under identical polymerization conditions and compared it to the E/VCH propagation rate constant; at 120 s, the *k_p_*PE, *k_p_*PP, *k*
*k_p_*
_P_E, *k_p_*_P_P, *k_p_*E, and *k_p_*VCH values are 4170, 250, 176, 21.53, 4266, 6410 and 93.6 L/mol·s, respectively [26,29,31], and the recommended moderate diffusion barriers are close to the real propagation rate constant in the early stages (Figure 8). In addition, the initial *k_p_*PE and *k_p_*E values of E homo- and E/VCH copolymerization are slightly different (4170 and 6410). Because the initial [C*]/[Zr] fraction of ethylene-VCH copolymerization is higher than that of ethylene homo-polymerization, and it is recommended that the active centers in these two system be of a different nature [18,25]. In contrast, the active center fraction value of E/P copolymerization is higher than E/VCH copolymerization, and the *k_p_*_P_E value of ethylene/propylene copolymerization is lower than E/VCH copolymerization, which means that these two copolymerization systems have different types of active centers. For example, in the ethylene homo- and E/VCH copolymerization, active centers may be loosely connected ion pairs with large *k_p_*E values. In contrast, active centers in the ethylene/propylene copolymerization active centers may be tightly linked ion pairs with low *k_p_*E.

To make comparisons between the propagation rate constant and polymer reaction time curves of polyethylene, ethylene/propylene, and ethylene/VCH copolymerization is more understandable; the *k_p_* versus polymer reaction time curves are shown in the comparable graph (Figure 9). The preliminary values obtained for the *k_p_* of ethylene/propylene copolymers are lower than those of the E/VCH copolymers because the E/P copolymers with a higher ratio of propylene (<20 mol%) remain nearly amorphous and may be liquefied in toluene, whereas E/VCH copolymers with lower VCH (mol 2%) are crystalline; this difference may be attributed to a more significant diffusion barrier in the E/VCH copolymer. The propagation rate constant of active canters developed initially in the E/VCH copolymer process tends to be greater than those formed later [25].

Previously reported E and P homo- and copolymerization with E/VCH copolymerization make the results more fascinating. It is well recognized that propylene is sterically larger than ethylene; a component of the homogeneous-metallocene precursor in the form of contact ion pairs might be incapable of promoting the propylene monomer for further coordination and might lead to its behavior as dormant or inactive in the homopolymerization system. Consequently, polypropylene’s [C*]/[Zr] ratio will be lower than polyethylene. In contrast, E/P copolymerization demonstrated a higher active center fraction than their homopolymers. This suggests that polypropylene can also reactivate the active sites that have been inactive during ethylene homopolymerization. Similarly, the active center fraction of E/VCH copolymers is higher than ethylene homo and lower ethylene/propylene co- polymers. This means that sterically VCH and other bicyclic dienes such as ENB and its derivatives activate the active site in the metallocene precursor that was inactive in the homopolymerization of E, but that the active site activation efficiency of VCH is less than α-olefins and ENB. Even though the cyclic olefins (VCH) have a less sterically linear-exocyclic_π_ bond and endocyclic_π_ bond, it is more difficult to synthesize than α-olefins such as propylene or 1-Hexen.

Despite the fact that the VCH is a sterically endocyclic_π_ bond, the complexes collapse or become weaker, and the inserting barriers are greater than that for P additions into the Zr–C. Finally, the introduction of VCH into the system led to a significant drop in inactivity. In addition, there is a steric influence on the VCH insertion due to the VCH endocyclic_π_ bond repulsion with the precatalyst ligand. In addition, endocyclic_π_ bond interaction with the precatalyst Zr metal is also possible (F). Furthermore, the lower active center [C*]/[Zr] percentage value of E/VCH copolymers than previously has been reported for ethylene/propylene copolymerization in both the early and stable stages can be related to a greater fraction of ZrCH_2_CH_3_-type dormant or inactive site in the former. Evidence supporting the existence of such dormant or inactive sites has been given in the literature [25,46,47]. As seen in Figure 2, ethylene-induced ZrCH_2_CH_3_ species produced in Zr-H can enter an inactive or dormant state through β-agostic contact with a Zr metal center and methyl hydrogen. As the electron deficit of the Zr center grows, the intensity of the β-agostic interaction increases as a function of time. By adding electron-donating substituents (such as phenyl and methyl etc.) on the catalytic ligand, the Zr center should have a reduced degree of electron deficiency and subsequently a lower proportion of inactive sites [38]. In the suggested mechanistic model (Figure 2), the catalytic precursor’s active sites with an ethylene/VCH propagating chain (A) may undergo β-H migration, leading to the construction of Cp2Zr–H species (B), which could then be converted to Cp2Zr–CH_2_–CH_3_ (c) by introducing ethylene into the Cp2Zr–H bond. Kissin Y.V. and his colleagues observed that they clarified the comonomer activation impact in the E with P and α-olefin polymer reactions employing commercialized heterogeneous synthetic polymeric Z–N catalysts in their model [1,11,14,15,16]. However, (C) has the ability to revert to a latent state and it is active for E and higher in the α-olefin polymer reaction. (D) Occurs as a result of strong β-agnostic interactions between hydrogen on methyl and Zr metal.

The presence of a greater proportion of these inactive or dormant sites in the E/VCH copolymerization is the primary cause for a smaller fraction of active caters in the copolymerization than in previously reported ethylene/propylene copolymerization investigations. It has been noted that the insertion of the VCH linear-exocyclic_π_ bond into the hydrogen–zirconium (H–Zr) bond during the E/VCH copolymerization can bypass the polymerization reaction, resulting in an increase in the active center fraction, which is higher than that of their homopolymerization counterparts. However, the VCH bypass polymerization reaction is slower than E/P, and subsequently results in a decrease in the active center fraction in VCH compared to E/P copolymers [29,46,48]

In the case of ethylene with cyclic diolefins copolymerization catalyzed by metallocene catalyst, the effect of cyclic olefin (VCH) on metallocene is very different, as explained in the discussion above. According to the investigation in the literature, ENB and VNB belong to the same class, and their effect was different on polymeric catalysts. Compared to ENB, VCH does not exhibit an electron-donating group at the linear-exocyclic_π_ bond that increases the electron deficiency of the edocyclic_π_ bond and shows the weakest activation effect on homogeneous metallocene catalysts for ethylene polymerization [31,34]. In addition, in the case of a low steric effect of the catalyst, the edocyclic_π_ bond of the VCH chelate with a catalyst metal atom leads to a strong deactivation effect and a lower active center fraction in the system (G).

In addition, VCH (CH_2_=CH–) insertion in the H–Zr bond can bypass the reactions in the initial stage more rapidly, decrease the dormant sites, and increase the active center fraction in the E/VCH copolymer system through Cp2Zr–(CH=CH–) (E). Peaks of edocyclic_π_ bonds in the constructed polymer chain verified the above discussion (Appendix A). However, the formation of active species (C) for the polymerization reaction and dormant or inactive sites (D) may need appropriate adjustments of zirconium metal and alkyl (Zr–iPr) moiety, which required sufficient space between the cation and anion. It is reasonable to believe that the [Cp2Zr-iPr] Cl-borate species with connected contact ion pairs is less likely to enter the inactive state than loosely linked ion pairs. It makes sense that the [Cp2Zr-iPr] Cl-borate species with loosely linked ion pairs is more easily transported into the dormant state than those with contact ion pairs. This assumption is confirmed by a significant increase in the [C*]/[Zr] ratio during the first 120–360 s of E/VCH copolymerizations (Table 1). As seen in Figure 4 and Figure 5, the incorporating rate of VCH is greater during the early polymerization period, and the [C*]/[Zr] approaches 35% and increases to 73% as the polymerization time increases. This suggests that catalytic species generated in the first 120–360 s with loosely connected ion pairs can insert more VCH than catalytic species activated later with contact ion pairs. In addition, loosely connected ion pairs supersede the active ethylene site to a great extent than contact ion pairs, with the result that molecular weight in the initial stage of the polymerization is lower than in the later stage. When all the results mentioned above are compared with previous reports, the E/NBE, E/VNB, and E/ENB copolymers, the E/VCH copolymer reaction with VCH comonomer, Mw, *R_p_*E, *R_p_*VCH, and the comonomer incorporation rate were significantly less than those of the E/NBE and E/ENB and E/VNB copolymers. This might be attributed to VCH special endocyclic_π_ bond. The endocyclic double bond in VCH may be able to engage with the active species remotely, leading to a smaller incorporation zone and a faster chain transfer process [34,35,46]. These findings suggest that minor differences in catalysts and comonomer topologies can influence polymerization behavior.

## 6. Conclusions

Symmetrical metallocene catalysts are useful polymeric catalysts for ethylene with α-olefin and cyclic olefin polymerizations. The copolymerization of E with vinyl addition polymerization of 4-vinyl-cyclohexene (VCH) was investigated using symmetrical *rac*-Et(Ind)_2_ZrCl_2_ combined with certain known borate/TIBA cocatalyst systems. To demonstrate the polymerizations’ living character, cyclic VCH with linear-exocyclic**_π_** and endocyclic**_π_** bonds produces polymers with a monomodal and narrow molecular weight distribution (MWD) regulated by reaction time. Secondly, the distribution of polymeric catalysts engaged in polymer chain synthesis and the nature of the dormant state are two of the most essential yet fundamentally unknown aspects. Precise research into the Ziegler–Natta polymeric catalyst has demonstrated that a comprehensive kinetic study based on acyl chloride selectively quenched the (Zr, Ti) metal–polymeric bond for assessing olefinic catalysts in the concepts of *k_p_*, *R_p_*, and [Zr]/[*C] ratio, and can be used to evaluate olefinic catalysts. The kinetic parameters *R_p_*E, *R_p_*VCH, *k_p_*E, and *k_p_*VCH, which are the apparent rates for metallocene-catalyzed E/VCH, *R_p_*E, and *k_p_*E values, are much more significant than *Rp*VCH and *k_p_*VCH at 120 s, *R_p_*E and *R_p_*VCH 39.63 and 0.78, and the *k_p_E* and *k_p_*VCH values are 6461 and 93 L/mol·s, respectively, and minor diffusion barriers are recommended in the early stages. When compared with P.E., VCH increases the *R_p_*E in the initial stage, which means that the exocyclic bond of VCH is more active at the initial level, and that the chain transfer reaction of cyclic internal π double is increased with the reaction time. The t_p_ versus *R_p_*, *k_p_*, and [Zr]/[C*] fraction count may be fitted to a model that provokes the deactivation of growing polymer chains. All of these properties need to use a specific kinetic mechanism that assumes a low activity of the developing polymer chains containing one E/VCH unit, the Zr–C_2_H_5_ group, in the E insertion process into the Zr–C bond. The copolymers obtained can be dissolved in conventional organic solvent and they have excellent thermal stability and crystalline temperature (Δ*H*_m_), their melting temperature (Tm)) varies from 109 to 126 °C, and Δ*H*_m_ varies from 80 to 128 (J/g). The Mw of the E/VCH copolymers was lower at the beginning of the reaction but gradually rose with t_p_. A narrow MWD which is nonetheless higher than two suggests numerous active [Zr]/[C*] sites in the catalytic system.

## Data Availability

Not applicable.

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
