# Peer review of "Methods for Predicting Ethylene/Cyclic Olefin Copolymerization Rates Promoted by Single-Site Metallocene: Kinetics Is the Key"

_polymers, 2022, doi:10.3390/polym14030459_

Round 1

Reviewer 1 Report

The paper submitted by Ali et al. deals with the investigation of ethylene/4-vinyl-cyclohexene copolymerization kinetics as a function of several parameters.

Even if this study is of interest, the quality of the presentation is quite poor. 1. They have used a lot of abbreviations without providing the full-name at the first use. 

2. In the abstract, the authors state that they obtained copolymers with a narrow molecular weight distribution (MWD). However, no such data appears in the manuscript. It is possible that the term "PD" from table 1 to be similar to the "dispersity" of samples but these values are very high. At this point, the authors must indicate which represent this abbreviation. 

3. The numbering of the references from the introduction section must be verified and corrected in order to have an homogeneous style. Numbers for ref 23 and 24 does not exist.

4. The materials section must be included in the manuscript.

5. The full name must be provided for all the abbreviations from table 1 and 2.

6. Revise caption for all the figures. (see other scientific articles)

7. The manuscript must be grammar and spell checked by a native English speaking. 

Author Response

Thank you very much for the deep insight and recommendation towards accepting the manuscript after  revision. 

Reviewer 2 Report

The authors present a manuscript regarding the possibility of prediction the copolymerization rate of ethylene and cyclic monomers.

Generally speaking hte paper is well written, al the parts are coherent and completed. The introduction is very nice and bring the reader cozily to the topic.

I have only some minor comments but I would really like the authors bring attention:

-the acronyms of the paper has to be resolved: please resolve them the first time you use them, as in line 1 of the introduction, NB, ENB etc. , do not be scared of resolve them more than once during the paper, a reader may also start reading from the middle.

-end of first parapraph and end of second paragraph of the introduction present citations formatted in a different way, please check

-the TOC is nice and colorfull but too mixed. all has to have a meaning, the arrow pointing up and down does they mean something?  the different colors of the three arrows exiting from the round bottom flask have a meaning or not? the position of the three structures means as the timeline in the bottom is pointing right that the structure on the left come first and then the middle and then right, correct? So, is visible that you have spent time doing this, is well done the backbone but revise all these issues please.

Author Response

Thank you very much for the deep insight and recommendation towards accepting the manuscript after minor revision. We have carefully revised the whole manuscript as per suggestions and recommendations.

Reviewer 3 Report

I am sorry for the delay, but I received the supplementary file. I agree with the publication of the article in the present form.

Author Response

We, authors, respect and appreciate the deep insight and recommendation towards accepting the manuscript. 

Round 2

Reviewer 1 Report

The authors have carried out some modifications but there are also other corrections which must be made:

  1. polidispersity (PD) is not the correct term; actually the recommended term is "dispersity" (Đ)
  2. Figure captions are still problematic. for example, the correct version for

- Figure 4: 1HNMR spectra of E/VCH copolymers

-Figure 5. Evolution of VCH and [Zr]/[C*] as a function of time

-Figure 6: Variation of VCH and RpVCH as a function of time

VCH values were provided 2 times, in fig 5 and fig 6. They should be deleted from one figure.

-Fig 7: Variation of kpE,  kpVCH and [Zr]/[C*] as a function of time

-Idem for Figure 8. 

The captions cannot be conclusions of the data presented in the figure.

Author Response

Thank you very much for the deep insight and recommendation towards accepting the manuscript after minor revision. 
